# Differentiation of Adipose-Derived Stem Cells into Vascular Smooth Muscle Cells for Tissue Engineering Applications

**DOI:** 10.3390/biomedicines9070797

**Published:** 2021-07-09

**Authors:** Alvaro Yogi, Marina Rukhlova, Claudie Charlebois, Ganghong Tian, Danica B. Stanimirovic, Maria J. Moreno

**Affiliations:** 1Human Health Therapeutics, National Research Council of Canada, 1200 Montreal Road, Ottawa, ON K1A0R6, Canada; Marina.Rukhlova@nrc-cnrc.gc.ca (M.R.); Claudie.Charlebois@nrc-cnrc.gc.ca (C.C.); Danica.Stanimirovic@nrc-cnrc.gc.ca (D.B.S.); 2Medical Devices Research Centre, National Research Council of Canada, 435 Ellice Ave, Winnipeg, MB R3B 1Y6, Canada; Ganghong.Tian@nrc-cnrc.gc.ca

**Keywords:** adipose stem cells, tissue engineering, differentiation, vascular smooth muscle, contraction

## Abstract

Synthetic grafts have been developed for vascular bypass surgery, however, the risks of thrombosis and neointimal hyperplasia still limit their use. Tissue engineering with the use of adipose-derived stem cells (ASCs) has shown promise in addressing these limitations. Here we further characterized and optimized the ASC differentiation into smooth muscle cells (VSMCs) induced by TGF-β and BMP-4. TGF-β and BMP-4 induced a time-dependent expression of SMC markers in ASC. Shortening the differentiation period from 7 to 4 days did not impair the functional property of contraction in these cells. Stability of the process was demonstrated by switching cells to regular growth media for up to 14 days. The role of IGFBP7, a downstream effector of TGF-β, was also examined. Finally, topographic and surface patterning of a substrate is recognized as a powerful tool for regulating cell differentiation. Here we provide evidence that a non-woven PET structure does not affect the differentiation of ASC. Taken together, our results indicate that VSMCs differentiated from ASCs are a suitable candidate to populate a PET-based vascular scaffolds. By employing an autologous source of cells we provide a novel alternative to address major issues that reduces long-term patency of currently vascular grafts.

## 1. Introduction

Cardiovascular diseases are the leading cause of hospitalization worldwide, often requiring bypass surgery for treatment of ischemic heart and peripheral vascular diseases. Native vessels remain the gold standard in revascularization procedures; however, due to age, disease, or prior usage for bypass surgery, autologous arteries or veins are not always available [1]. Synthetic grafts using various materials have been developed and have demonstrated acceptable long-term patency for the replacement of large (>8 mm) and medium (6–8 mm) diameter vessels; however, for small diameter conduits (<6 mm), low compliance, thrombogenicity and lack of vaso-mechanical properties limits their use [2,3]. For this reason, alternative strategies are being investigated such as the grafting of vascular cells on biodegradable [4] or non-biodegradable [5] scaffolds to create artificial vascular substitutes that mimic the biochemical and biomechanical properties of native arteries.

Using a melt-blown technique we produced non-woven polyethylene terephthalate (PET) small-diameter vascular grafts with porosity and fiber diameter control and compliance match of that of native arteries [6]. To recreate the anatomical structure of a vessel, human endothelial and smooth muscle cell lines were respectively grafted in the luminal and abluminal side of the tubular scaffold using a proprietary cell-seeding device [7]. With this method, a high cell-seeding efficiency, viability, uniformity and reproducibility were achieved [7]. The next step for the clinical translation of the tissue-engineered vascular grafts is the generation of a cellularized scaffold with autologous cells that meets the following criteria: (i) rapid biofabrication, (ii) stable phenotype of the grafted cells, (iii) compatible with the topography and chemical properties of the biomaterials used as scaffold, and (iv) suitable for preclinical evaluation in animal studies, such as the porcine model.

Smooth muscle cells are the main component of the blood vessel tunica media and play an important role in maintaining homeostasis and the mechanical properties of the vasculature. The recreation of this middle layer represents a significant challenge in tissue engineering due to the difficulty to harvest autologous smooth muscle cells and their rapid loss of phenotype during in vitro culture [8]. To overcome these issues, stem cells have been investigated for their inherent ability to differentiate into diverse cell types, including vascular lineages [9,10]. Adipose tissue is an ideal source of stem cells (adipose-derived stem cells; ASCs) as they can be easily and repeatedly harvested using minimally invasive procedures [11,12,13,14,15,16,17]. Differentiation of vascular smooth muscle cells (VSMC) from ASCs can be achieved by culture supplementation with members of the transforming growth factor beta (TGF-β) superfamily [18]. TGF-β has been shown to coordinate α-smooth muscle actin (α-SMA) expression and the assembly of α-actin and actin filaments in stem cells [19]. Furthermore, targeting signaling pathways induced by TGF-β or other members of its family such as bone morphogenic proteins (BMP) inhibits the expression of VSMC-specific contractile proteins in stem cells, highlighting their critical role in the differentiation process [20,21,22,23].

In this study, we investigated important aspects of ASC differentiation into a VSMC phenotype for the translatability of the generated vascular graft: (i) the temporal expression of VSMC markers after long (7 days) and short (4 days) exposure of human-derived ASCs to TGF-β and BMP4; (ii) the downstream effector(s) involved in this process; (iii) the long-term stability of the VSMC phenotype; (iv) the compatibility of the process with the topographic and chemical properties of the PET biomaterials; and (v) the reproducibility of the process using porcine-derived ASCs.

## 2. Materials and Methods

### 2.1. Isolation and Culture of Human and Porcine Adipose-Derived Stem Cells

Human mediastinal adipose tissue was collected from five patients during open-chest cardiac surgery at St. Boniface General Hospital. Informed consent was obtained from all the patients prior to collection of the adipose tissue and the study was previously approved by the Human Ethic Committee (IBD2008.02, approval date 19 November 2008).

Pig subcutaneous adipose tissue was collected from the abdomen region under a general anesthesia. The animals used in this study received humane care in compliance with the Guide to the Care and Use of Experimental Animals formulated by the Canadian Council on Animal Care. The Animal Care Committee of National Research Council Canada approved the experimental protocols of this study. ASCs were then isolated from the adipose tissues according to the method developed by Zuk and colleagues [17] with some modifications. In brief, the excised adipose tissue was washed extensively with phosphate-buffered saline (PBS, HyClone, Logan, UT, USA) to remove contaminating debris and blood cells. The adipose tissue was minced and digested with collagenase I (2 mg/mL, Worthington Biochemical, Lakewood, NJ, USA) at 37 °C for 20–30 min. Collagenase activity was then neutralized by a complete medium. The complete medium contains Dulbecco’s Modified Eagle Medium (DMEM, HyClone, Logan, UT, USA) with 15% fetal bovine serum (HyClone, Logan, UT, USA), 50 μg/mL streptomycin (HyClone, Logan, UT, USA) and 50 IU/mL penicillin (HyClone, Logan, UT, USA). The digested adipose tissue was filtered with a 100 μm and then with a 25 μm nylon membrane (BD Falcon, Bedford, MA, USA), to eliminate the undigested fragments. The cellular suspension was centrifuged at 300× *g* for 10 min. The cell pellets were re-suspended in the complete medium and cultivated for 48 h at 37 °C in 5% CO_2_. Adherent cells were disassociated with 0.25% trypsin (Life Technologies, Burlington, ON, Canada) and collected for subsequent studies.

### 2.2. Differentiation of ASC

ASCs were differentiated in DMEM containing 1% FBS, 5 ng/mL of TGF-β (R&D Systems, Minneapolis, MN, USA) and 2.5 ng/mL of BMP-4 (R&D Systems, Minneapolis, MN, USA) for up to 4 or 7 days To investigate the long-term retention of the differentiated phenotype, following the 4 or 7 days differentiation protocol, cells were either transferred to DMEM containing 10% FBS or maintained in differentiation medium for an additional 3, 7 or 14 days (for the 7 days differentiation protocol) or 3, 11 or 18 days (for the 4 days differentiation protocol).

### 2.3. Western Blot Analysis

ASCs were lysed at day 0, 1, 2, 3, 4, 5, 6 or 7 following the differentiation and non-differentiation protocols and proteins were separated by SDS-PAGE and transferred onto a nitrocellulose membrane. Membranes were incubated overnight at 4 °C with the following primary antibodies: anti-α-SMA (1:10,000, rabbit, monoclonal, cat# 04-1094, Millipore, Billerica, MA, USA), anti-SM22α (1:500, mouse, monoclonal, cat# 28811, Abcam, Cambridge, UK), anti-caldesmon (1:500, mouse, monoclonal, cat# MAB3576, Millipore, Billerica, MA, USA), anti-IGFBP7 (1:300, goat, polyclonal, cat# AF1334, R&D Systems, Minneapolis, MN, USA). Washed membranes were incubated with horseradish peroxidase-conjugated secondary antibody (1:5000‒1:50,000, cat# A9044, cat# AP106P, cat# AP187P Sigma-Aldrich, St. Louis, MS, USA and Millipore, Billerica, MA, USA). Immunoreactive proteins were detected by chemiluminescence. Signals were revealed by chemiluminescence, visualized by autoradiography and quantified densitometrically. β-Actin (1:50,000, mouse, monoclonal, cat# A2228, Sigma-Aldrich, St. Louis, MO, USA) expression was used as a housekeeping protein.

### 2.4. Immunofluorescent Staining

Immunofluorescence was performed on methanol-fixed cells initially blocked with 10% normal goat serum in PBS supplemented with 0.1% Triton-X for 1 h at RT. Slides were incubated with the following primary antibodies: anti-α-SMA (1:100, rabbit, monoclonal, cat# 04-1094, Millipore, Billerica, MA, USA), anti-SM22α (1:50, mouse, monoclonal, cat# 28811, Abcam, Cambridge, UK), anti-caldesmon (1:50, mouse, monoclonal, cat# MAB3576, Millipore, Billerica, MA, USA), anti-IGFBP7 (1:100, goat, polyclonal, cat# AF1334, R&D Systems, Minneapolis, MN, USA). After washing three times in PBS, slides were incubated with Alexa 568 IgG (1:300, goat, polyclonal, cat# A-11004, Thermo Fisher Scientific, Rockford, IL, USA). Glass slides were then rinsed in PBS and milli-Q water and coverslipped in fluorescent mounting medium (Dako Diagnostics, Glostrup, Denmark) spiked with Hoechst (DAPI, 1:1000, cat# D9542, Sigma-Aldrich, St. Louis, MO, USA). Sections were visualized under the Olympus 1 × 2 UCB fluorescent microscope and images were captured using in vivo v.3.2.2 software (MediaCybernetics, Rockville, MD, USA).

### 2.5. Collagen Lattice Contraction Assay

Collagen solution was produced by mixing acidic-soluble type I collagen (Life Technologies, Grand Island, NB, USA) 3 mg/mL, PBS 10x and water according to the manufacturer’s instructions. Collagen solution was mixed with cell suspension in serum-free medium, neutralized with 1M NaOH and plated in six-well cell culture cluster (Corning, Tewksbury, MS, USA) and gelled at room temperature for 20 min. The final concentration of collagen was 4.0 mg/m with a cell population of 300,000 cells per well. Two milliliters of serum-free DMEM was then poured onto the gel to prevent the surface from dehydrating. The gel was detached from each well and left floating. Contraction was induced by KCl at a final concentration of 60 mM. Surface area of gel samples was measured at detachment (time 0) and after 2, 4 and 6 h. Gel area was calculated on acquired images by computer-assisted morphometric analysis (Scion Image). The contraction of the gel was expressed as percentage of initial lattice area following the formula: A2/A1 × 100, where A1 is the initial gel area and A2 the area at the observed interval. Three culture plates were used for each experimental group.

### 2.6. Statistical Analysis

Results are expressed as the mean ± SEM. ANOVA followed by Newman–Keuls’ post-test was used to compare multiple groups. A *p*-value of less than 0.05 was considered statistically significant.

## 3. Results

### 3.1. TGF-β- and BMP-4-Induced Differentiation of ASCs into Smooth Muscle Cell

Human ASCs (hASC), previously characterized for their adipogenic, myogenic and osteogenic differentiation potential [24], were stimulated with differentiation media containing TGF-β (5 ng/mL) and BMP-4 (2.5 ng/mL) for up to 7 days. Western blot analysis, indicated a time-dependent upregulation of α-SMA, caldesmon and SM22α expression, in all cases reaching a plateau at around day 4–6 (Figure 1A). α-SMA expression was increased after 3 days of differentiation, while SM22α and caldesmon expression was only significantly increased after 4 and 5 days, respectively. Immunofluorescence staining further confirmed the differential expression of SMC markers upon incubation of hASCs with TGF-β and BMP-4 for 4 and 7 days (Figure 1B). To assess whether the hASC-derived smooth muscle-like cells exhibit contractile properties, cells were embedded in collagen lattices and the dimension of the collagen lattices was determined at different time points after depolarization with 60 mM KCl. As shown in Figure 1C,D, collagen gel lattices embedded with either hASCs differentiated for 7 days or with aortic VSMC, used as positive control, displayed a time-dependent reduction in size. Depolarization with 60 mM KCl did not induce contraction in gels seeded with undifferentiated hASCs.

Due to the similarities in the cardiovascular system between humans and pigs, porcine models are frequently used for the pre-clinical evaluation of tissue-engineered vascular grafts. We investigated whether the differentiation protocol used with human ASCs could be applied to ASCs of porcine origin. Appendix A shows that TGF-β and BMP-4 induce the expression of SMC molecular markers in porcine ASCs in a similar manner as they do in human ASCs. This was demonstrated by Western blot (Appendix A) and immunofluorescence. (Appendix A) Taken together, our results indicate that ASCs from human or porcine origin can be differentiated into a SMC-like phenotype with a combination of TGF-β and BMP-4.

### 3.2. Role of TGF-β and BMP-4 in ASC Differentiation

To better understand the individual roles of TGF-β and BMP-4 in ASC differentiation, protein expression profiles of molecular markers of SMC were analyzed by Western blot and immunofluorescence in human ASCs after single treatment with either TGF-β or BMP-4. TGF-β alone significantly increased the expression of α-SMA after 4 days of treatment and the levels continuously increased for up to 7 days (Figure 2A). However, these levels were significantly lower than those induced by the combination of TGF-β and BMP-4. TGF-β also induced the expression of caldesmon (Figure 2B) and SM22α (Figure 2C), although significant levels were only observed at late time points, 5 and 6 days, respectively. BMP-4 alone induced a significant increase in caldesmon expression after 6 days of treatment (Figure 2B) but did not induce changes in either α-SMA or SM22α expression during the 7 days of treatment. (Figure 2C) Immunofluorescence images further confirmed the Western blot findings (Figure 2D).

### 3.3. Role of IGFBP7 in ASC Differentiation

We have previously demonstrated that TGF-β induces the expression and secretion of IGFBP7 in U87MG glioblastoma cells [25]. Considering that IGFBP7 has also been shown to induce α-SMA expression in different cell types [26] we questioned whether IGFBP7 plays a role in stem cell differentiation. Figure 3A shows a time-dependent increase in IGFBP7 expression after exposure of human ASCs to differentiation media containing TGF-β and BMP-4. The maximum effect was observed after 2 days, but the expression levels remained significantly increased for up to 7 days of exposure. An increase in IGFBP7 expression was also observed in ASCs exposed individually to either TGF-β or BMP-4, although these levels were significantly lower than those observed in cells exposed to the combination of both (Figure 3A). This indicates a possible synergistic pathway between these two agents in mediating IGFBP7 expression. These observations were further confirmed by immunofluorescence (Figure 3B).

We next examined whether IGFBP7 alone or in combination with BMP-4 could induce the differentiation of ASCs into SMC-like cells. Human ASCs were maintained in media supplemented with IGFBP7 at equimolar concentration (30.5 ng/mL) of TGF-β, BMP-4 (2.5 ng/mL) or both and examined for 7 days for α-SMA, caldesmon and SM22α expression by Western blot and immunofluorescence. Interestingly, we observed that the replacement of TGF-β with IGFBP7 resulted in a delayed expression of α-SMA (Figure 4A), caldesmon (Figure 4B) and SM22α (Figure 4C), with significant expression being observed only after 5 or 6 days of differentiation compared to non-differentiated hASCs. In the presence of IGFBP7 alone no changes in α-SMA, caldesmon and SM22α expression was observed during the 7 days of treatment. Immunofluorescence staining further confirms the results obtained by Western blot (Figure 4D).

### 3.4. Stability and Long-Term Differentiation of ASC

To determine the long-term stability of SMCs derived from human ASC, cells treated with differentiation media (TGF-β and BMP-4) for 7 days were either placed back into regular cell growth media (DMEM supplemented with 10% FBS) or maintained in differentiation media for an extra 3, 7 or 14 days. Western blot analysis indicate that hASCs kept in differentiation media containing TGF-β and BMP-4 for up to 21 days (Figure 5A) as well as those switched to DMEM supplemented with 10% FBS (Figure 5B) retained the expression of α-SMA, caldesmon and SM22α at similar levels to those observed at 7 days. These results were further confirmed by immunofluorescence of α-SMA (Figure 5B). In addition to the expression of molecular markers, the ability to contract in response to a depolarizing challenge (60 mM KCl) was also observed in hASCs treated with differentiation media for 4 days (Figure 5C). Moreover, the reduction of collagen gel lattice size was similar in the cells maintained in differentiation media containing TGF-β and BMP-4 when compared to those switched to DMEM supplemented with 10% FBS for an extra 3, 7 or 14 days (Figure 5C).

The stability of the differentiated phenotype was also tested in porcine adipose-derived stem cells. Appendix A shows that cells differentiated for 7 days and then switched to DMEM containing 10% FBS still maintained high levels of α-SMA, caldesmon and SM22α (Appendix A) expression. These results indicate that differentiated ASCs obtained from porcine sources can maintain their acquired VSMC phenotype for up to 21 days.

### 3.5. Short-Term Differentiation of ASC into SMC

Based on our initial findings indicating that significantly increased levels of the SMC markers were observed before 7 days of differentiation, we tested whether cells differentiated for a shorter period of time would have the potential to be used in the development of an autologous cell seeded vascular graft. To this end, we differentiated human ASCs with TGF-β and BMP-4 for 4 days. Cells were then either placed in DMEM medium supplemented with 10% FBS or maintained in the differentiation media for an additional 3, 7 or 14 days. Immunofluorescence images show α-SMA and caldesmon expression in cells maintained in medium containing TGF-β and BMP-4 for 4 days (Figure 6A) Furthermore, the expression of these markers was also present when cells were maintained in DMEM medium supplemented with 10% FBS for up to 14 days. After 4 days of differentiation, human ASCs were able to contract in response to 60 mM KCl (Figure 6B). More importantly, KCl-mediated contraction was similar when differentiated hASCs were placed back in DMEM/10% FBS for 3, 7 and 14 days compared to those maintained in differentiation media. Taken together, these data suggest that hASC differentiation can be achieved within 4 days. Considering that the length of production of a vascular graft is a major issue, reducing the differentiation period would have a beneficial impact when it comes to clinical applications.

### 3.6. ASC Differentiation in Synthetic Surface

We next questioned whether the synthetic surface in PET scaffolds developed in our group for vascular tissue engineering applications influence the differentiation of ASCs mediated by TGF-β and BMP-4. To address this issue, human ASCs were seeded and then treated with differentiation media in the PET surface rather than the regular polystyrene surface and cellular markers of VSMCs determined at day 4 and 7. Figure 7A,B shows that TGF-β and BMP-4 time-dependently increased α-SMA, SM22α and caldesmon expression in hASCs grown in PET surface. We also observed that long-term differentiation and stability of the molecular markers induced by TGF-β and BMP-4 were also maintained when cells were treated for protocols that lasted up to 21 days (Figure 7A,B). Taken together, these results demonstrate that the differentiation process can be performed in the synthetic surface of the scaffold and highlight the potential use of hASCs in the generation of a fully mature autologous vascular graft.

## 4. Discussion

Our study demonstrates that ASCs obtained from human and pig adipose tissue can be differentiated into smooth muscle cells with a mature contractile phenotype upon exposure to TGF-β and BMP-4. In particular, we show that with this protocol, differentiation can be obtained in as little as 4 days. In addition, we also found that one of the TGF-β downstream effectors, IGFBP7, along with BMP-4, plays a role in this process. Another major finding from our study was that the mature VMSC phenotype was stable for over 21 days and was not affected by the seeding of cells on a PET non-woven polymeric fibrous material specifically designed to serve as a vascular scaffold with compliance properties similar to those of native vessels [6].

One of the major limitations of the engineered vascular graft technology for clinical applications is the lengthy production time required prior to implantation. For example, a promising approach based on completely autologous grafts generated from patient’s skin fibroblasts (Cytograft Inc., Novato, CA, USA) requires approximately 6–10 months for complete biofabrication. This time-consuming process is often not compatible for use in patients that need rapid intervention. Therefore, current efforts are aimed at shortening the production time while still ensuring the optimal properties of the cellularized tissue-engineered vascular grafts.

Adipose tissue provides a readily available and relatively easily obtainable source of autologous pluripotent stem cells [11,14,16]. ASCs have been shown to be amenable for differentiation into VSMCs using different cell culture protocols including long exposure (3–6 weeks) to heparin [27], or by shorter incubation (7 days) in culture media containing either TGF-β (10 ng/mL) [28] or TGF-β plus BMP-4 (5 ng/mL and 2.5 ng/mL, respectively) [20]. Obtaining a vascular graft with properties that closely resemble native vessels is critical for implantation success. VSMCs account for the majority of the extracellular matrix and ultimately define the mechanical properties of scaffolds. Therefore, for tissue engineering applications, reducing the constraints of a long production process by obtaining VSMCs with a mature phenotype in minimal time is ideal. Here, we found that, similarly to previously reported results, TGF-β plus BMP-4 synergistically promoted a time-dependent increase in the expression of early (α-SMA) and late (caldesmon and SM22α) VSMC markers. More importantly, this VSMC phenotype was fully achieved in 4 days and was stable up to 21 days whether the ASCs were kept in differentiation media or switched after 4 days to regular growth media. The stability of the VSMCs achieved in this study is especially important considering reports indicating that the acquired phenotype might be only transient under certain differentiation protocols [29].

The TGF-β family consists of over 30 members, including the TGF-βs and bone morphogenetic proteins that are involved in numerous cell processes. TGF-β is a critical cytokine in the differentiation of stem cells into VSMCs [20,21,22,23]. Nonetheless, the exact signaling cascades and key downstream effectors involved in this effect remain to be determined. It has been demonstrated that TGF-β induces the expression of IGFBP7 through SMAD2-dependent mechanisms [25,30]. In accordance, here, we demonstrate that IGFBP7 expression is increased in ASCs treated with differentiation media. This effect was synergistically induced by TGF-β and BMP-4 since the expression of IGFBP7 was significantly reduced in ASCs exposed to each individual factor. This synergism can be due to the fact that both BMP-4 and TGF-β activate Smad-dependent pathways that converge into the nucleus to modulate the transcription of numerous target genes. It has been shown that that IGFBP7 knockdown blunts TGF-β-induced α-SMA expression in renal proximal epithelial cells [30]. Interestingly, in our study, IGBP7 alone failed to induce the expression of α-SMA in ASCs as well as that of caldesmon and SM22α. Similarly, BMP-4 individually did not stimulate the expression of α-SMA and SM22α in ASCs and only slightly induced the expression of caldesmon at a late time point (6–7 days). However, in combination, BMP-4 and IGFBP7 significantly increased the expression of molecular markers of SMC differentiation, albeit with lower levels when compared to differentiation media containing TGF-β and BMP-4. Taken together, our results indicate that IGFBP7 plays an important role in mediating the differentiation of ASCs into VSMCs induced by TGF-β and BMP-4, although IGFBP7-independent pathways also seem to be involved in this process and will be the subject of future studies.

The interaction between cells and the topographic and chemical surfaces of biomaterials is a powerful tool for modulating cell differentiation, phenotype and function for regenerative medicine applications [31]. Using a melt-blowing process, we previously developed a PET scaffold for small diameter vascular tissue engineering applications [6]. By varying the flow rate of the molten polymer, this fabrication method allows the control of the fiber diameter distribution. In addition, the pore size and compliance of the scaffold was modulated by the number and directionality of the fiber webs forming the structure. We observed that structures with smallest fiber diameter (1–6 µm) and narrow pore size range (1–20 µm) and porosity 70% were the most suitable for the growth of endothelial and vascular smooth muscle cells. The tubular scaffolds produced with this structure exhibited compliance properties (~8.4 × 10^−2^ % mmHg^−1^) closely matching those of native vessels (5‒8 × 10^−2^ % mmHg^−1^) [7]. In the present study, we demonstrate that the PET scaffold is also biocompatible with ASCs and supports the differentiation of ASCs into VSMCs using TGF-β and BMP-4. Overall expression of molecular markers and stability were similar to what was observed in cell culture dishes, suggesting the feasibility of using PET-based scaffolds for vascular tissue engineering.

Although small- and medium-sized animal models are suitable for basic screening and concept evaluation, large animal models are generally required for preclinical safety and efficacy testing in vascular tissue engineering applications. Pigs are often the preferred pre-clinical model for cardiovascular applications considering that the distribution of blood supply by the coronary artery system is almost identical to that of humans [32]. In our study we successfully validated the differentiation of porcine-derived ASCs into SMC, indicating that our protocol is suitable to be carried out on a pre-clinical model. Comparing the characteristics of porcine and human-derived ASCs is especially important since it has been shown that in some instances the capacity and functionality of differentiated cells obtained from different sources may not be similar [33].

In conclusion, our results indicate that differentiation of ASCs into VSMCs can be obtained within 4 days of exposure to TGF-β and BMP-4. We also showed that not only molecular markers of VSMCs were present but also the ability to contract in response to a depolarizing agent. Furthermore, the phenotype obtained was stable for up to 21 days even in the absence of the differentiating agents. The cells obtained here are a suitable candidate to populate a PET-based vascular scaffold, and the protocol is suitable for the use in ASCs from human and porcine sources. By employing an autologous source of cells with a reduced and stable process of differentiation we provide an alternative to the development of a fully mature vascular graft that addresses major issues hindering the deployment of this technology in the clinic.

## Figures and Tables

**Figure 1 biomedicines-09-00797-f001:**
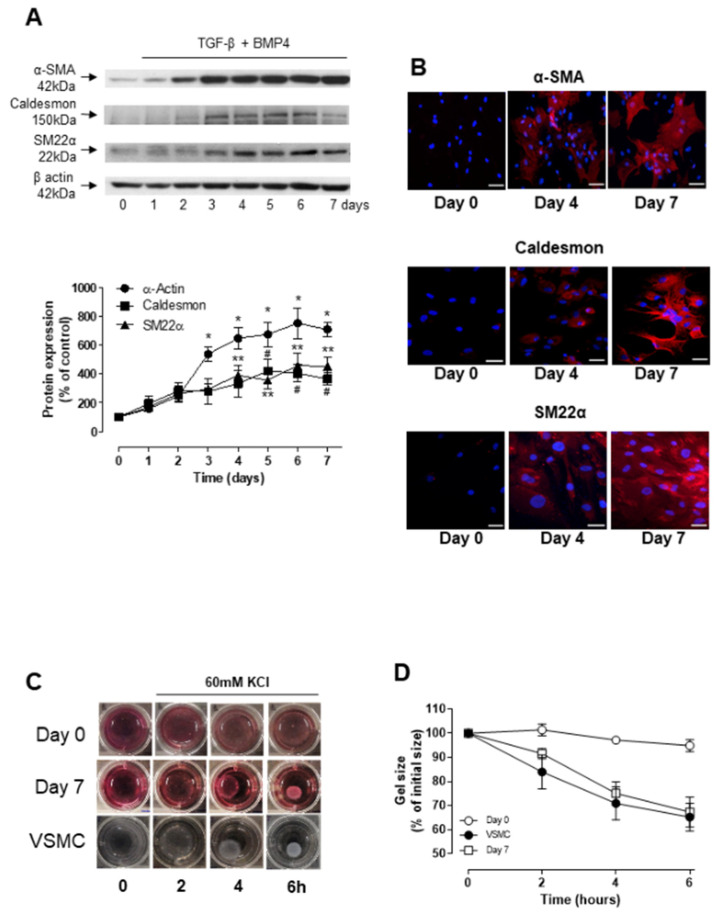
TGF-β and BMP-4 induces differentiation of human ASCs into SMC-like phenotype. (**A**) Top are representative immunoblots for αSMA, caldesmon and SM22α expression. Corresponding line graph demonstrate the time-course (0 to 7 days) effect of TGF-β and BMP-4 on αSMA, caldesmon and SM22α expression. * *p* < 0.05 vs. αSMA expression at day 0, ** *p* < 0.05 vs. SM22α expression at day 0, # *p* < 0.05 vs. caldesmon expression at day 0. (**B**) Fluorescence microscopy was also used to detect αSMA, caldesmon and SM22α expression in hASCs at day 0, 4 and 7 of TGF-β and BMP-4 stimulation. Scale bar 100 µm. (**C**,**D**) Contractile activity of non-differentiated hASCs (Day 0) and hASCs differentiated for 7 days with TGF-β and BMP-4 embedded into collagen gel matrices. The collagen gel matrices were cultured in the serum-free medium for up to 6 h and gel contraction was photographed at the indicated time points using a digital camera. Corresponding line graph demonstrating the time-course (0 to 6 h) effect of 60 mM of KCl on the area of the gel lattices relative to its original size. * *p* < 0.05 vs. non-differentiated hASCs at 6 h, ** *p* < 0.05 vs. non-differentiated hASCs at 6 h. All results are mean ± SEM of 5 experiments.

**Figure 2 biomedicines-09-00797-f002:**
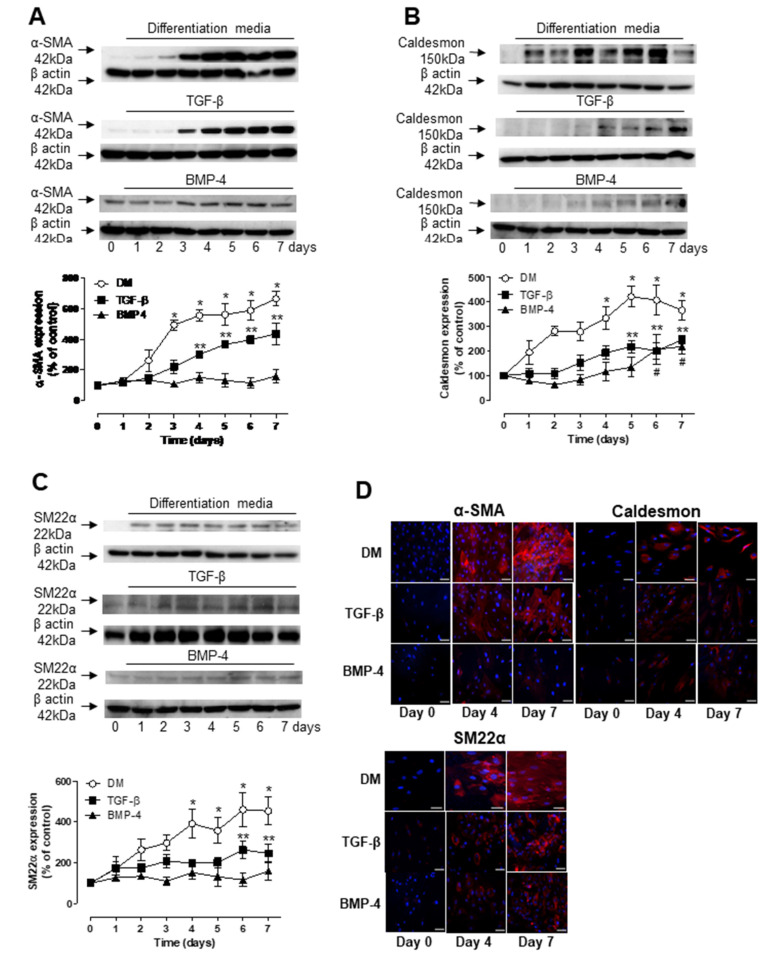
TGF-β and BMP-4 have a synergistic effect on human ASC differentiation. hASCs were stimulated either with TGF-β, BMP-4 or in combination (DM) for up to 7 days and αSMA (**A**), caldesmon (**B**) and SM22α (**C**) expression were evaluated by Western blot and fluorescence microscopy (**D**). Scale bar 100 µm. Top are representative immunoblots for αSMA, caldesmon and SM22α expression. Corresponding line graph demonstrating the time-course (0 to 7 days) effect of TGF-β, BMP-4 or DM on αSMA, caldesmon and SM22α expression. Results are mean ± SEM of 5 experiments. * *p* < 0.05 vs. day 0 of stimulation with DM; ** *p* < 0.05 vs. day 0 of stimulation with TGF-β; # *p* < 0.05 vs. day 0 of stimulation with BMP-4.

**Figure 3 biomedicines-09-00797-f003:**
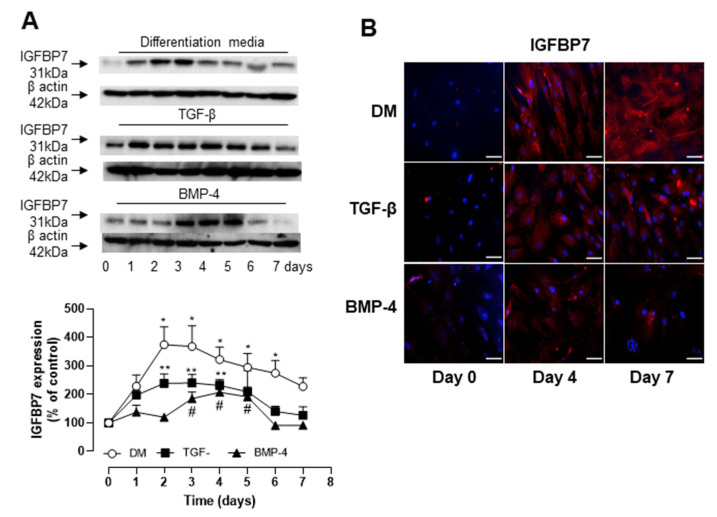
IGFBP7 expression is increased in human ASCs going through differentiation with TGF-β and BMP-4. (**A**) Top is representative immunoblot for IGFBP7 expression. Corresponding line graph demonstrating the time-course (0 to 7 days) effect of TGF-β and BMP-4 on IGFBP7 expression. TGF-β and BMP-4 synergistically induce the expression of IGFBP7 on hASC. hASCs were stimulated with TGF-β, BMP-4 or combination of TGF-β and BMP-4 (DM) for up to 7 days and IGFBP7 expression was evaluated by Western blot. Top are representative immunoblots for IGFBP7 expression. Corresponding bar graph demonstrating the time-course (0 to 7 days) effect of TGF-β, BMP-4 or DM on IGFBP7 expression. (**B**) Fluorescence microscopy was also used to detect TGF-β, BMP-4 or DM-induced IGFBP7 expression on day 0, 4 and 7. Scale bar 100µm. Results are mean ± SEM of 5 experiments. * *p* < 0.05 vs. day 0 of stimulation with DM; ** *p* < 0.05 vs. day 0 of stimulation with TGF-β; # *p* < 0.05 vs. day 0 of stimulation with BMP-4.

**Figure 4 biomedicines-09-00797-f004:**
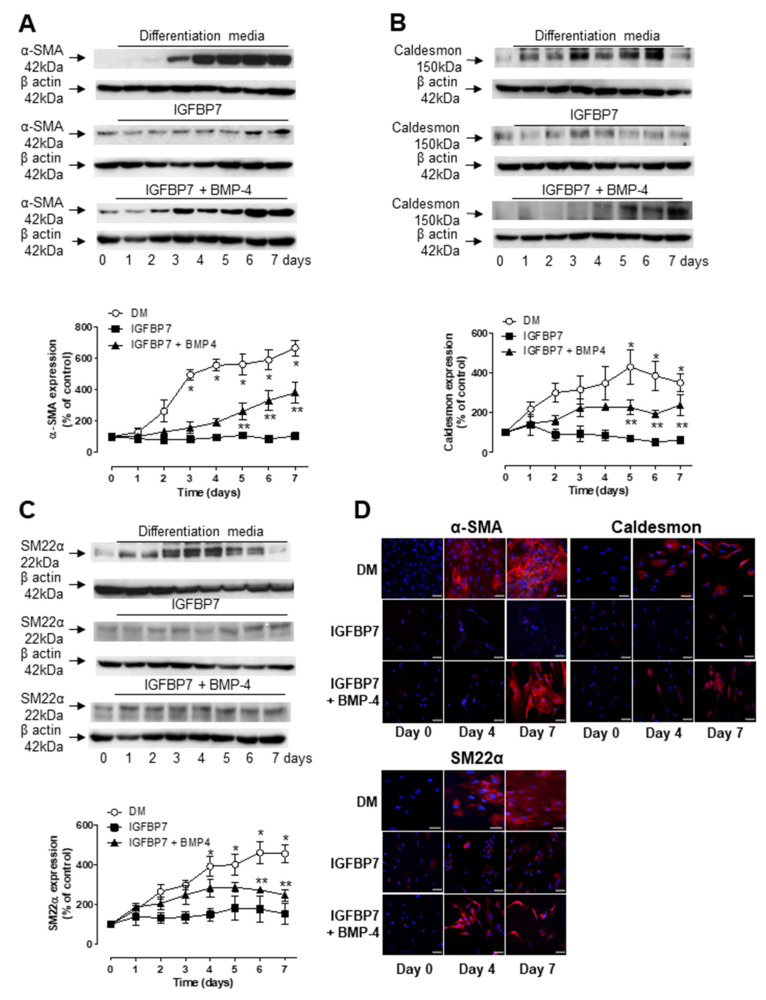
Combination of IGFBP7 and BMP-4 but not IGFBP7 alone induced expression of molecular markers of SMC. Human ASCs were stimulated with differentiation media containing TGF-β and BMP-4 (DM), IGFBP7 or combination of IGFBP7 and BMP-4 for up to 7 days. Top are representative immunoblots for αSMA (**A**), caldesmon (**B**) and SM22α (**C**) expression. Corresponding line graph demonstrating the time-course (0 to 7 days) effect of DM, IGFBP7 or IGFBP7 and BMP-4 on αSMA, caldesmon and SM22α expression. (**D**) Western blot analysis was further confirmed by fluorescence microscopy to detect αSMA, caldesmon and SM22α expression at day 0, 4 and 7 following stimulation with DM, IGFBP7 or IGFBP7 and BMP-4. Scale bar 100 µm. Results are mean ± SEM of 5 experiments. * *p* < 0.05 vs. day 0 of stimulation with DM; ** *p* < 0.05 vs. day 0 of stimulation with or IGFBP7 and BMP-4.

**Figure 5 biomedicines-09-00797-f005:**
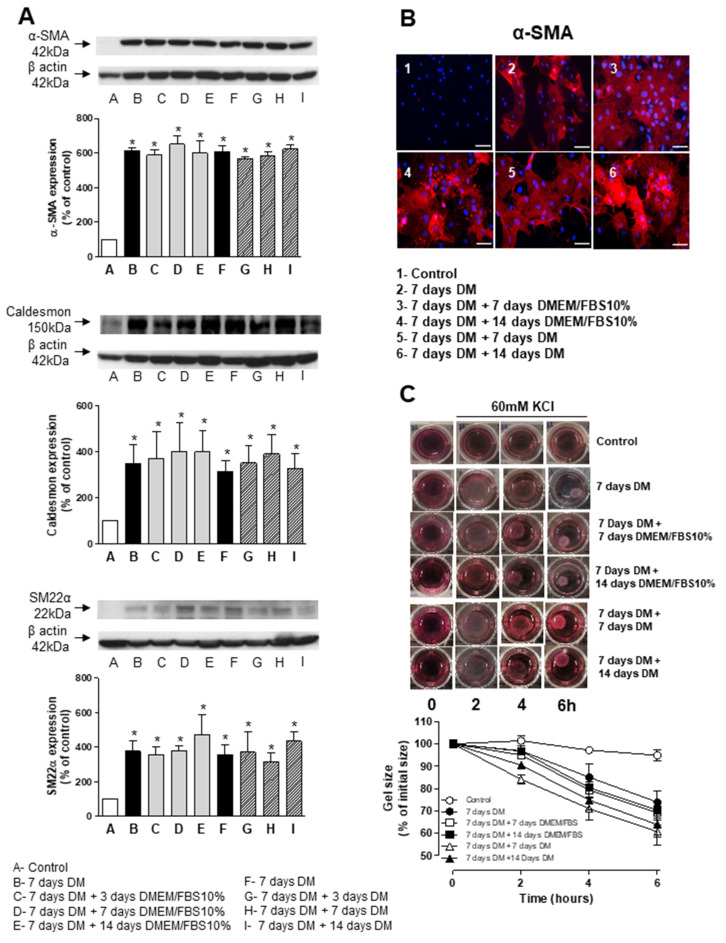
TGF-β- and BMP-4-induced differentiation of human ASCs is stable for up to 21 days. (**A**) Top are representative immunoblots for αSMA, caldesmon and SM22α expression. Corresponding bar graph demonstrating the stability of differentiation when cells were differentiated for 7 days and then either maintained in differentiation media containing TGF-β and BMP-4 or transferred to DMEM supplemented with FBS for indicated times. (**B**) Fluorescence microscopy was also used to detect αSMA in hASsC differentiated for 7 days and either kept in DM or switched to DMEM supplemented with FBS for indicated times. Scale bar 100 µm. (**C**) Contractile activity of non-differentiated hASCs and hASCs differentiated for 7 days and then either maintained in differentiation media containing TGF-β and BMP-4 or transferred to DMEM supplemented with FBS for indicated times. The collagen gel matrices were cultured in the serum-free medium for up to 6 h and gel contraction was photographed at the indicated time points using a digital camera. Corresponding line graph demonstrating the time-course (0 to 6 h) effect of 60 mM of KCl on the area of the gel lattices relative to its original size. Results are mean ± SEM of 5 experiments. * *p* < 0.05 vs. non-differentiated ASC.

**Figure 6 biomedicines-09-00797-f006:**
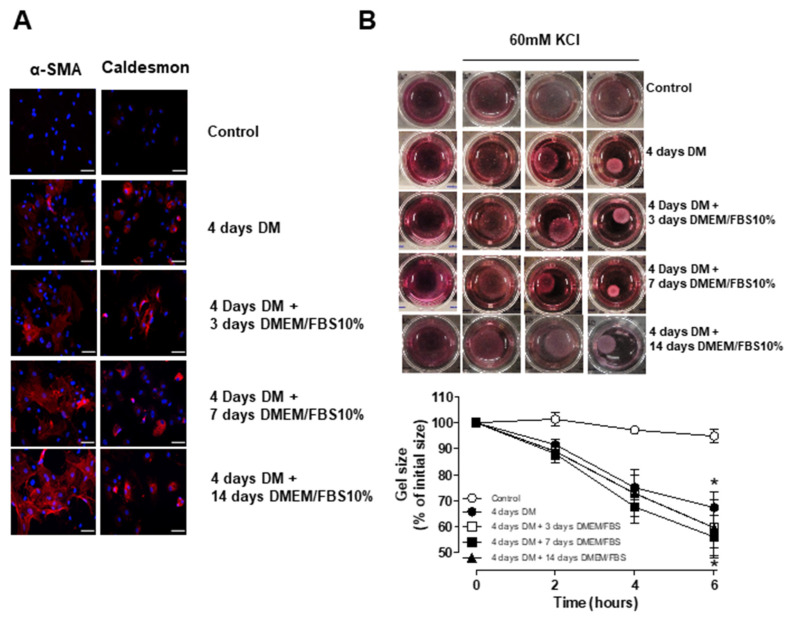
Human ASCs can be differentiated into SMC after 4 days of treatment with TGF-β and BMP-4. (**A**) Rather than 7 days, cells were differentiated for 4 days and then transferred to DMEM containing FBS for indicated times. αSMA and caldesmon expression was detected by fluorescence microscopy. Scale bar 100 µm. (**B**) Contractile activity of non-differentiated hASCs and hASCs differentiated for 4 days and then transferred to DMEM supplemented with FBS for indicated times. The collagen gel matrices were cultured in the serum-free medium for up to 6 h and gel contraction was photographed at the indicated time points using a digital camera. Corresponding line graph demonstrating the time-course (0 to 6 h) effect of 60 mM of KCl on the area of the gel lattices relative to its original size. Bar graphs indicate the area of the gel lattices relative to its original size after 6 h. Results are mean ± SEM of 5 experiments. * *p* < 0.05 vs. non-differentiated ASC.

**Figure 7 biomedicines-09-00797-f007:**
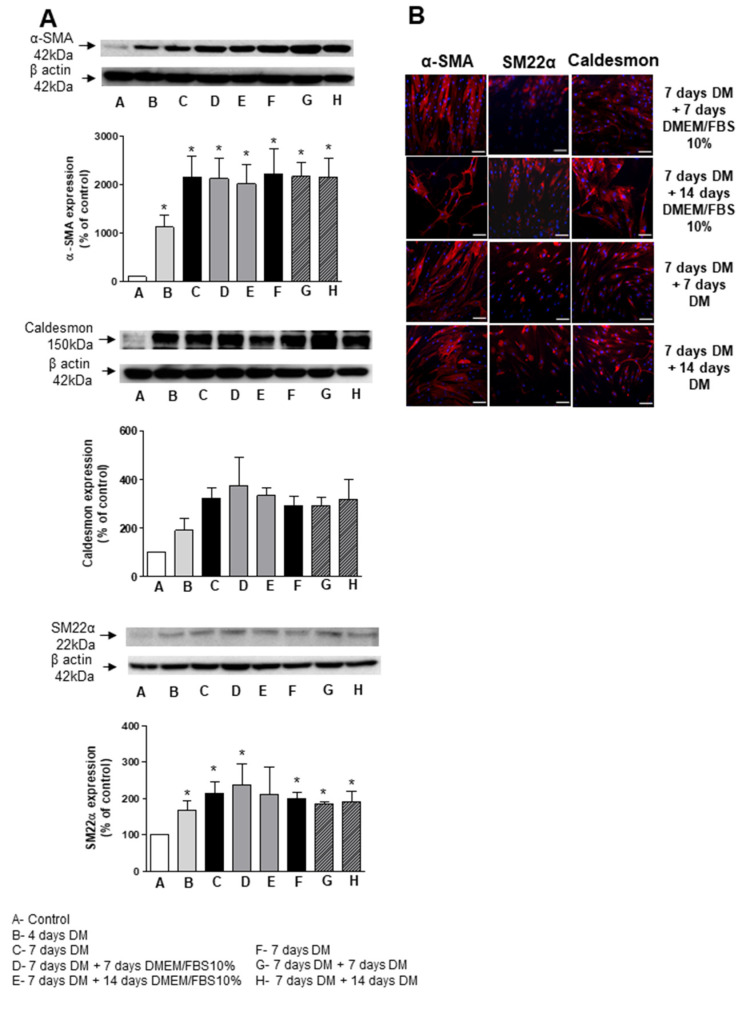
PET structures do not affect the differentiation of human ASCs into SMC mediated by TGF-β and BMP-4. Long-term stability of differentiation was evaluated on hASCs grown in PET surface. Cells were differentiated for 7 days and then either maintained in differentiation media containing TGF-β and BMP-4 or transferred to DMEM supplemented with FBS for indicated times. (**A**) Top are representative immunoblots for αSMA, SM22α and caldesmon expression. Corresponding bar graph demonstrate protein expression of αSMA and caldesmon evaluated by Western blot. (**B**) Fluorescence microscopy was also used to detect αSMA, SM22α and caldesmon expression in ASCs grown in PET surface at indicated time points. Scale bar 100 µm. Results are mean ± SEM of 5 experiments. * *p* < 0.05 vs. non-differentiated ASC.

## Data Availability

The data that support the findings of this study are available from the corresponding author, upon request.

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
