# Peer review of "Differentiation of Adipose-Derived Stem Cells into Vascular Smooth Muscle Cells for Tissue Engineering Applications"

_biomedicines, 2021, doi:10.3390/biomedicines9070797_

Round 1

Reviewer 1 Report

In the publication entitled "Differentiation of adipose-derived stem cells into vascular smooth muscle cells for tissue engineering applications" the authors fully described the interactions of adipose-derived cells with PET-based scaffolds developed for cardiovascular tissue engineering. There's no doubt, the data and methods provided will be useful for the researchers in this field. 
However, its needed to introduce material fabrication and characterisation (e.g. SEM, water contact angle) parts for better understanding, as well. Since these data will be added, the manuscript, undoubtedly will be of great interest for the researchers. 

Author Response

In the publication entitled "Differentiation of adipose-derived stem cells into vascular smooth muscle cells for tissue engineering applications" the authors fully described the interactions of adipose-derived cells with PET-based scaffolds developed for cardiovascular tissue engineering. There's no doubt, the data and methods provided will be useful for the researchers in this field.

However, its needed to introduce material fabrication and characterisation (e.g. SEM, water contact angle) parts for better understanding, as well. Since these data will be added, the manuscript, undoubtedly will be of great interest for the researchers.

We would like to thank the reviewer for all the comments, specifically for pointing out the lack of sufficient information on the fibrous scaffolds used in this study. The non-woven PET fiber structures used in the present study were previously described in two publications (Moreno et al., JBMR, 2011; and Mohebbi-Kalhori et al, J Tissue Eng Regen Med, 2011). Briefly, a melt-blowing process was used to produce nonwoven PET fiber structures in which the fiber diameter was modified by varying the flow rate of the molten polymer. In addition, the pore size and compliance of the scaffold was modulated by the number and directionality of the fiber webs forming the structure. In this study we used the structure E described in Moreno et al, JBMR, 2011. The tubular scaffolds produced with this structure had 6cm length, internal diameter 6.0mm, wall thickness 150±7 mm, fibre diameter range 1.57–15.8 mm, pore size range 1-20 mm and porosity 70%. These scaffolds exhibited compliance properties (~8.4 x 10-2 % mmHg-1) closely matching those of native vessels (5-8 x 10-2 % mmHg-1). They also had the biocompatibility requirements for the seeding of endothelial and vascular smooth muscle cells in the luminal and abluminal sides of the tubular scaffolds, respectively (Mohebbi-Kalhori et al., J Tissue Eng Regen Med, 2011). SME images and protocols depicting the biomechanical characterization of these scaffolds are fully described in Moreno et al., JBMR, 2011.

This information has now been added to the paragraph 5 of the discussion section in the manuscript.

Reviewer 2 Report

Well organized paper with plenty of data that supports their hypothesis. The data and scientific rigor are clear and concise. It was not clearly stated but was assuming that the work in the paper was done using Human ASC except for the data in the supplemental figures which was done with porcine ASC. Do the pig ASC behave in a similar manner as the human ASC when grown on the PET scaffold? The contractility assay that was used was interesting and one I have not seen before. Other than KCl, would this assay work using other  contractile drugs such as norepinephrine? 

Author Response

Well organized paper with plenty of data that supports their hypothesis. The data and scientific rigor are clear and concise. It was not clearly stated but was assuming that the work in the paper was done using Human ASC except for the data in the supplemental figures which was done with porcine ASC.

We apologize to the reviewer for any confusion regarding the origin of the adipose-derived stem cell in our study. Yes, as the reviewer indicated, all the data presented in the paper was done using human ASC except for the data in the supplemental figures, which was done with porcine ASC. We have modified the manuscript accordingly to clearly indicate whether the cells being used in each experiment were of human or porcine origin.

Do the pig ASC behave in a similar manner as the human ASC when grown on the PET scaffold?

Considering that expression of smooth muscle cells markers induced by TGF-β and BMP-4 were similar in cells from human and porcine origin, and that the PET surface had no impact on the differentiation of human ASC, we expect that pig ASC would acquire similar phenotype as human ASC in our non-woven structures. This analysis will be the subject of future studies since the pre-clinical assessment of vascular graft performance is normally evaluated in pigs.

The contractility assay that was used was interesting and one I have not seen before. Other than KCl, would this assay work using other contractile drugs such as norepinephrine?

Contraction is a hallmark of smooth muscle cell phenotype and the collagen lattice contractile assay is a well established method to investigate this phenomenon. This assay is based on the finding that cell-populated collagen hydrogels contract over time in a predictable, and consistent manner in response to certain stimuli. Not only a depolarizing agent such as KCl can be used, but also numerous vasoactive agents such as endothelin 1, angiotensin II, arginine, vasopressin, acetylcholine and norepinephrine were shown to reduce the size to the collagen lattice in multiple studies. Further characterization of the contractile phenotype of the ASC-derived VSMCs will be investigated in future studies.

Reviewer 3 Report

The authors produced PET vascular grafts as artificial scaffolds and loaded endothelial and smooth muscle cell lines on the scaffold to mimic the vessel. To prove the translatability of this vessel graft, they investigated the important aspects of the ACS differentiation into a VSMC phenotype using this graft. Although the authors have been done lots of experiments and characterization in this manuscript, the writing style makes it pretty difficult to understand the innovative points and the real problem to be solved in this work, especially in the abstract part. 

  1. I don't find any characterization and fabrication methods of the PET biomaterials device. If the author claim they investigated the topographic and chemical properties of the PET biomaterials, the SEM images of the topographic should be provided. 
  2. The author should explain why they choose 7 days as long exposure and  4 days as a short exposure period? is there any specific physiological significance? why didn't use 14 days to investigate the long-term effect?
  3. The scale bar information in all fluorescent imaging is missing.

Author Response

The authors produced PET vascular grafts as artificial scaffolds and loaded endothelial and smooth muscle cell lines on the scaffold to mimic the vessel. To prove the translatability of this vessel graft, they investigated the important aspects of the ACS differentiation into a VSMC phenotype using this graft. Although the authors have been done lots of experiments and characterization in this manuscript, the writing style makes it pretty difficult to understand the innovative points and the real problem to be solved in this work, especially in the abstract part.

We appreciate the feedback from the reviewer on the need to highlight the innovative aspects of our study. Our main objective was to develop a fast and stable smooth muscle phenotype using autologous adipose-derive stem cells for the production of fully cellularized vascular grafts that could address the current limitations of purely synthetic grafts (eg. thrombogenicity, compliance, biomechanical response, etc). To the best of our knowledge we have for the first time: i) provided a detailed analysis of the temporal expression of smooth muscle cell markers on adipose-derived stem cells differentiated with TGF-β and BMP-4 , ii) identified the minimum time required for the ASC differentiation into a mature contractile smooth muscle phenotype, iii) demonstrated the individual role of TGF-β, BMP-4 and IGFBP7, a downstream signaling effector of TGF-β, in this process, iv) demonstrate biocompatibility and feasibility of the differentiation protocol in PET fibrous scaffolds, previously developed for the generation of tissue-engineered vascular graft with compliance properties mimicking those of native vessels, and v) evaluate the differentiation protocol in porcine-derived adipose stem cells as an important step for preclinical assessment of the vascular graft performance in pigs before implantation in humans. We hope this clarifies the innovative aspects of the present study. We have made modifications in the abstract and body of the manuscript (Discussion section, paragraph 1) to emphasize the novelty of our findings and how we expect that this will further advance the field to obtain a fully mature vascular graft based on autologous cells.

1.I don't find any characterization and fabrication methods of the PET biomaterials device. If the author claim they investigated the topographic and chemical properties of the PET biomaterials, the SEM images of the topographic should be provided.

We would like to thank the reviewer for this comment. The non-woven PET fiber structures used in this study were previously described in two publications (Moreno et al., JBMR, 2011, and Mohebbi-Kalhori et al, J Tissue Eng Regen Med, 2011). The non-woven PET fiber structures used in the present study were previously described in two publications (Moreno et al., JBMR, 2011; and Mohebbi-Kalhori et al, J Tissue Eng Regen Med, 2011). Briefly, a melt-blowing process was used to produce nonwoven PET fiber structures in which the fiber diameter was modified by varying the flow rate of the molten polymer. The pore size and compliance of the scaffold was modulated by the number and directionality of the fiber webs forming the structure. In this study we used the structure E described in Moreno et al, JBMR, 2011. The tubular scaffolds produced with this structure had 6cm length, internal diameter 6.0mm, wall thickness 150±7 mm, fibre diameter range 1.57–15.8 mm, pore size range 1-20 mm and porosity 70%. These scaffolds exhibited compliance properties (~8.4 x 10-2 % mmHg-1) closely matching those of native vessels (5-8 x 10-2 % mmHg-1). They also had the biocompatibility requirements for the seeding of endothelial and vascular smooth muscle cells in the luminal and abluminal sides of the tubular scaffolds, respectively (Mohebbi-Kalhori et al., J Tissue Eng Regen Med, 2011). SME images and protocols depicting the biomechanical characterization of these scaffolds are fully described in Moreno et al., JBMR, 2011.

This information has now been added to the paragraph 5 of the discussion section in the manuscript.

2.The author should explain why they choose 7 days as long exposure and 4 days as a short exposure period? is there any specific physiological significance? why didn't use 14 days to investigate the long-term effect?

The issue of the differentiation timeline is of great importance to the field and we would like to thank the reviewer for raising this concern. Considering that cardiovascular patients may require fast intervention after a critical insult, our goal was to determine the minimum time required to generate a fully matured vascular graft ready for implantation. In the literature there are different protocols describing the differentiation of adipose-derived stem cells into vascular smooth muscle cells. For instance, Salem and cols. (PMID: 24044001) differentiated adipose-derived stem cells with medium containing 100U/mL of heparin for up to six weeks. Chen and Dean demonstrated that this differentiation can be reduced to 7 days when cells are grown in the presence of 10ng/ml TGF-β1 (PMID: 30686947). In the present study we sought to further characterize the protocol described by Wang and cols. that obtained vascular smooth muscle cells by incubating adipose-derived stem cells with a mixture of TGF-β and BMP-4 (5ng/mL and 2.5ng/mL, respectively) for 7 days (PMID: 19895205). Our first step was to characterize the temporal effects of TGF-β and BMP-4 in the expression of molecular markers of smooth muscle cell over the course of 7 days. Our results indicated that the expression of smooth muscle cells markers reached a plateau after 4 days of exposure to TGF-β and BMP-4. We then asked whether the differentiation period could be shortened. In a subsequent study we demonstrated that the cells exposed to TGF-β and BMP-4 for only 4 days, not only expressed all the smooth muscle cell markers but that they were also capable of contracting in response to a depolarizing challenge. Based on our previous results (PMID: 21948700), after seeding cells into a scaffold, the cells have to proliferate and invade the 3D structure, so we estimated the need for 1 or two extra weeks growing in the bioreactor under pulsatile flow conditions to obtain a fully mature vascular graft. Therefore, in our study we investigated the stability of the differentiation process during this time frame. We demonstrated that the smooth muscle cell phenotype was stable for 14 additional days after being transferred into normal media. This long-term stability of the smooth muscle cell phenotype highlights the suitability of using autologous adipose-stem cells combined with the current differentiation protocol for the development of the media layer of tissue-engineered vascular grafts.

3.The scale bar information in all fluorescent imaging is missing.

We would like to thank the reviewer for pointing out the lack of scale bar in the fluorescent images. A scale bar 100 µm has been included in the images in the revised manuscript.

Round 2

Reviewer 1 Report

In this version of the manuscript, the authors took into account all the comments of the reviewers.This article may be accepted for publication.
This article may be accepted for publication.

Reviewer 3 Report

In the revised version the authors have addressed my comments satisfactorily. I recommend the paper for publication.